Identification and expression profile analysis of the SnRK2 gene family in cucumber

Wan Zilong
Luo Shilei
Zhang Zeyu
Liu Zeci
Qiao Yali
Gao Xueqin
Yu Jihua
Zhang Guobin zhanggb@gsau.edu.cn
1 Gansu Agricultural University, State Key Laboratory of Arid Land Crop Science , Lanzhou , China
2 College of Horticulture, Gansu Agricultural University , Lanzhou , China
Ganopoulos Ioannis
Electronic publication date: 2022 Sep 21
Publication date: 2022
Volume: 10
Electronic Location ID: e13994
Received 2022 Apr 29; Accepted 2022 Aug 11
Copyright: ©2022 Wan et al.
Copyright year: 2022
Copyright holder: Wan et al.
License: This is an open access article distributed under the terms of the Creative Commons Attribution License, which permits unrestricted use, distribution, reproduction and adaptation in any medium and for any purpose provided that it is properly attributed. For attribution, the original author(s), title, publication source (PeerJ) and either DOI or URL of the article must be cited.
License URL: https://creativecommons.org/licenses/by/4.0/

Keywords: Bioinformatics analysis, Cucumber, SnRK2 gene family, Gene expression

Funding: Director Foundation of Key Laboratory of Arid Habitat Crop Science of Gansu Province, China GHSJ-2020-Z3 Education science and technology innovation project of Gansu Province 2021CYZC-45 This research was supported by Director Foundation of Key Laboratory of Arid Habitat Crop Science of Gansu Province, China (GHSJ-2020-Z3) and Education science and technology innovation project of Gansu Province (2021CYZC-45). The funders had no role in study design, data collection and analysis, decision to publish, or preparation of the manuscript.

==============================
The sucrose non-fermenting-1-related protein kinase 2 (SnRK2) is a plant-specific type of serine/threonine protein kinase that plays an important role in the physiological regulation of stress. The objective of this study was to identify and analyze the members of the SnRK2 gene family in cucumber and lay a foundation for further exploration of the mechanism of CsSnRK2 resistance to stress. Here, 12 SnRK2 genes were isolated from cucumber and distributed on five chromosomes, phylogenetic clustering divided these into three well-supported clades. In addition, collinearity analysis showed that the CsSnRK2 gene family underwent purifying selection pressure during evolution. CsSnRK2 genes of the same group have similar exons and conserved motifs, and intron length may be a specific imprint for the evolutionary amplification of the CsSnRK2 gene family. By predicting cis elements in the promoter, we found that the promoter region of CsSnRK2 gene members had various cis-regulatory elements in response to hormones and stress. Relative expression analysis showed that CsSnRK2.11 (group II) and CsSnRK2.12 (group III) were strongly induced by ABA, NaCl and PEG stress; whereas CsSnRK2.2 (group III) was not activated by any treatment. The response of group I CsSnRK2 to ABA, NaCl and PEG was weak. Furthermore, protein interaction prediction showed that multiple CsSnRK2 proteins interacted with four proteins including protein phosphatase 2C (PP2C), and it is speculated that the CsSnRK2 genes may also an independent role as a third messenger in the ABA signaling pathway. This study provides a reference for analyzing the potential function of CsSnRK2 genes in the future research.

Introduction

Biological and abiotic stressors greatly inhibit plant growth and development, severely affecting crop yield and quality, and, under various stress conditions, plants form complex response mechanisms. Among them, protein phosphorylation catalyzed by protein kinase is an important mechanism of signal transduction in plant cells (Huang et al., 2012; Yu, An & Li, 2014; Wang et al., 2001). The SnRK is a serine/threonine protein kinase that is widely present in plants. It can be divided into three subfamilies (SnRK1, SnRK2, and SnRK3) according to the conservation of the kinase activity domain (Halford, Boulyz & Thomas, 2000; Kulik et al., 2011; Umezawa et al., 2013). The abscisic acid (ABA) signaling pathway is a key pathway enabling plants to cope with adversity, and SnRK2 participates in the ABA signal transduction, forming the ABA-PYR-PP2C-SnRK2-downstream transcription factor coupling signaling pathway (Hrabak et al., 2003). In ABA deficiency, PP2C interacts with SnRK2 to inhibit the activity of related kinases, and the signaling pathway is closed. When ABA is present, its receptor binds to the relevant hormones and acts on PP2C, releasing the inhibition of phosphatase SnRK2 and opening the signaling pathway (Cheng et al., 2017; Sheard & Zheng, 2009). At present, owing to the important role of SnRK2 in different stress conditions, many plant-related SnRK2 gene families have been identified and studied; including Arabidopsis thaliana (10 SnRK2s) (Saha et al., 2014), rice (10 SnRK2s) (Saha et al., 2014), maize (11 SnRK2s) (Huai et al., 2008), soybean (22 SnRK2s) (Wei et al., 2017), cotton (20 SnRK2s) (Liu et al., 2017), wheat (10 SnRK2s) (Zhang et al., 2016), sorghum (10 SnRK2s) (Li-Bin et al., 2010), tomato (7 SnRK2s) (Sun et al., 2011), mungbean (8 SnRK2s) (Fatima et al., 2020), grapevine (8 SnRK2s) (Liu et al., 2016), pak-choi (13 SnRK2s) (Huang et al., 2015), and Brassica napus (114 SnRK2s) (Zhu et al., 2020).

The expression of the SnRK2 gene derived from wheat is induced by ABA, drought, and other stresses (Holappa & Walker-Simmons, 1995; Liu et al., 2017). In Arabidopsis, except for AtSnRK2.9, the other nine SnRK2 genes were found to be induced by mannitol and NaCl (Boudsocq, Barbier-Brygoo & Lauriere, 2004). AtSnRK2.2-3 and SnRK2.6 are core regulators of ABA signal transduction, and play key roles in coping with water stress and in controlling seed development and dormancy (Fujita, Yoshida & Yamaguchi-Shinozaki, 2013; Yasunari et al., 2009). All members of the SnRK2 gene family in rice can be induced by NaCl, among which SAPK8, SAPK9, and SAPK10 are induced by ABA (Kobayashi et al., 2004). Among the 11 ZmSnRK2 genes in maize, ZmSnRK2.3/6 w as strongly induced by NaCl treatment, while most of the other genes were weakly induced by salt stress; ZmSnRK2.3/7 was strongly induced by low-temperature treatment (Huai et al., 2008). Plants overexpressing ZmSnRK2.11 showed reduced sensitivity to salt and drought under salt and drought stress treatments (Zhang et al., 2015). Compared with normal growing plants, the overexpression of the TaSNRK2.4 gene prolonged the growth cycle and increased the yield of wheat plants. Additionally, Arabidopsis plants have an increased resistance to drought, high salinity and low temperatures (Mao et al., 2010). Recent studies have shown that SnRK2 regulates a variety of plant growth and development processes in the ABA signaling pathway, including stomatal opening and closing, seed dormancy and germination, and the flowering cycle (Fujii & Zhu, 2009; Wang et al., 2013). For example, the SnRK2 gene in grapes is involved in the stomatal regulation of osmotic stress (Boneh et al., 2012). In summary, different SnRK2 genes are involved in the response of plants to various stressors.

Cucumber is an important vegetable cash crop, that has significant production and economic value worldwide (Wen et al., 2016). Currently, drought and other adverse stresses conditions have serious effects on the growth, development, photosynthesis, yield, and quality of cucumber. Transcriptome and transgenic analyses have shown that many cucumber genes are abnormally expressed under salt and drought stress (Du et al., 2021; Ouzounidou et al., 2016; Wang et al., 2012). Therefore, improving the cucumber yield and quality by increasing their resistance to stress is a particularly important initiative. In this study, 12 cucumber SnRK2 gene family members were identified, as well as systematically and bioinformatically analyzed. In addition, we analyzed the relative gene expression of cucumber under normal environmental conditions and several stressors. This study provides a theoretical basis of the functions of this gene family for further studies.

Material and Methods

Plant materials

Cucumber seeds L306 (Cucumis sativus L, “L306” cultivar) were used in this experiment. The seeds were disinfected with 5% sodium hypochlorite and then germinated until the two true leaves were completely expanded. The seedlings were cultured in Yamazaki cucumber nutrient solution at 25 °C/18 °C (day/night), 250 µmol.m−2 s−1 light and 70% relative humidity in an artificial climate chamber. Samples were collected at three true leaf stages. This experiment set up four treatments: 50 µM ABA, 100 µM ABA, 200 mM NaCl and 10% PEG. In each treatment, a total of 20 seedling samples of similar size were treated. Five sampling time points were set for different treatments, and the leaves of four seedlings were collected at each time point (0, 3, 6, 12, and 24 h). RNA was extracted after cutting and mixing, and three CT values were measured as repetition.

Identification of CsSnRK2 genes

To screen candidate SnRK2 genes in cucumber, BLASTp was used to compare the protein sequences of 10 SnRK2s of Arabidopsis with that of cucumbers using an online database (http://cucurbitgenomics.org/). Set the BLASTp e-value to 1e−10 (Altschul, 1997). The redundancy of the results was removed manually. The hidden Markov model (HMM) of the kinase domain (PF00069.25) was obtained from the Pfam database (http://pfam.xfam.org/), and the cucumber genome database was further searched by HMMER 3.0 software. TBtools was also used to screen candidate genes; Pfam (http://pfam.xfam.org/search#tabview=tab1) and NCBI-CDD (https://www.ncbi.nlm.nih.gov/cdd/) databases were used for validation and candidate genes without SnRK2 domains (registration number: PF00069.25) were manually deleted (Zhou et al., 2018).

Chromosome localization and basic information analysis

For the distribution on chromosomes, the gff3 file of ”9930” V3 version of China long was downloaded. SnRK2 genome location information was obtained by screening. It was named CsSnRK2 s according to its distribution on chromosomes. The chromosome map of the gene was drawn by MG2C (http://mg2c.iask.in/mg2c_v2.0/). WoLF PSORT (https://wolfpsort.hgc.jp/) was used to predict protein subcellular localization (Xiong et al., 2015). The number of amino acids as well as the molecular weights, isoelectric points, and other physical and chemical information of CsSnRK2 gene were analyzed on ExPaSy website (https://web.expasy.org/protparam/) (Liu et al., 2017). The secondary structure of the cucumber SnRK2 protein was predicted by using the PRABI website (https://npsa-prabi.ibcp.fr/cgi-bin/npsa_automat.pl?page=npsa_sopma. html).

Collinearity analysis of CsSnRK2 genes

Collinearity analysis using MCScanx provided by TBtools software. Cucumber genome was compared with Arabidopsis genome, maize genome and rice genome by BLASTp, and the search threshold was set as e-value <1e−5. Other default parameters have not been modified. The results of genome-wide BLASTp were used to calculate all possible collinear pairs between chromosomes. Using TBtools to map the collinear pairs of SnRK2 genes in cucumber, Arabidopsis, maize and rice (Chen et al., 2020). The protein sequences of the gene pairs were extracted for multiple sequence alignment, and then the alignment results and the CDS sequences of the gene pairs were used to calculate the non-synonymous/synonymous (dN/dS) values of duplicate gene pairs using PAL2NAL (http://www.bork.embl.de/pal2nal/index.cgi?) (Wang et al., 2021).

Phylogenetic, structure and conserved motif analysis

The phylogenetic trees of SnRK2 gene family of cucumber, A.thaliana, rice, maize, sugar beet, tomato, grapevine and pak-choi were constructed by using MEGA 11 software with ClustalW methods to align multiple sequence (Kumar, Stecher & Tamura, 2016). The bootstrap values of 1000 replicates were calculated at each node. The minimum neighbor method was selected to represent the phylogenetic tree. Then, the phylogenetic tree was visualized using the iTOL online tool (http://itol.embl.de/) (Ivica & Peer, 2016). The CDs and whole genome sequences of 12 cucumber SnRK2 genes were uploaded to GSDS (http://gsds.cbi.pku.edu.cn/) to visualize the positions of introns and exons and untranslated regions (Hu et al., 2014). We analyzed the conserved motifs of gene by MEME (http://meme-suite.org/tools/meme). The maximum motif number was set to 10, and the other parameters are the default values (Bailey et al., 2009).

Analysis of cis-acting elements, tissue-specific expression and protein interaction networks

TBtools were used to extract the genomic sequence 2,000 bp upstream of the start codon (ATG) of SnRK2 genes in cucumber. Then, the 2000 bp sequences of 12 CsSnRK2 were submitted to PlantCARE (http://bioinformatics.psb.ugent.be/webtools/plantcare/html/) for prediction. The RNA-Seq data of cucumber in different tissues and organs (tendril-base, tendril, root, leaf, stems, ovary-unfertilized, ovary-fertilized, ovary) were obtained from the cucumber genome data using the registration number PRJNA80169 (http://cucurbitgenomics.org/). The data were converted by log2 method, and TBtools software was used to draw the heat map of CsSnRK2 genes expression. Using 12 CsSnRK2 protein sequences as targets, the protein-protein interaction network was predicted by using STRING website (https://string-db.org/cgi).

RNA extraction and Real-Time PCR (qRT-PCR)

Total RNA was isolated from collected samples using a Plant RNA Extraction kit (Tiangen, China). cDNA was synthesized with fastking cDNA dispersion RT supermaxs Kit (Tiangen, China) and 2uL RNA was used as template. Design of primers using CDs sequences of genes. The SYBR Green kit (Tiangen, China) was used for fluorescence quantitative analysis. The volume of reaction system was 20 uL, which contained 2 uL cDNA solution, 10 uL 2*SuperReal PreMix Plus, 0.6 uL of 10 uM forward and reverse primers, 0.4 uL 50*ROX Reference Dye and 6.4 uL of distilled deionized water. qRT-PCR analysis using LightCycler® 480 II real-time fluorescent quantitative PCR instrument. Each qRT-PCR reaction was carried out with three technical replicates. Cucumber actin (DQ115883) was used as internal control. The amplification primers and internal reference primers are shown in Table 1. Amplification conditions were as follows: 95 °C for 15 min, and 40 cycles of 95 °C for 10 s and 60 °C for 30 s. The relative expressions levels of the SnRK2 genes were calculated using the 2−ΔΔCt method (Livak & Schmittgen, 2002). SPSS 20.0 software was used for one-way ANOVA, and Duncan method was used for significance test (p < 0.05). Using origin 9.0 software to complete the histogram of relevant expression, and the values of the last drawn graph are the average values of three repetitions.

Table 1 Primer sequences for qRT-PCR.

Gene Name	Forward Primer Sequence (5′–3′)	Reverse Primer Sequence (5′–3′)	
CsSnRK2.1	GCAGATCATCGAGGAAGCGTCAG	CAAGTCCAGTTCAGCATCCGTCTC	
CsSnRK2.2	TTATGCACGACAGTGACCGCTATG	GTCTCTCATCAACCTGGCAACCC	
CsSnRK2.3	TTGGTTGGAGCATACCCGTTTGAG	GCGAGAAAGAAGGTTGCGACATTC	
CsSnRK2.4	TCATCCTTGCTGCATTCGAGACC	CCACACGACCATACATCTGCCATC	
CsSnRK2.5	TTAGAAGAAAGCAAAGTCCC	ATGTGGTCGGAAGTGAGAAG	
CsSnRK2.6	CGTTGGTACTCCGGCATACATAGC	CCACACGACCATACATCTGCCATC	
CsSnRK2.7	TAGTTGGAGGGCAAGGGCAATTTG	CCAAGGCACAAACAAAGTCACCAC	
CsSnRK2.8	GGATCGCTCCGCACTTACTGTTG	ACCGAAGTTTCCAGAGCCAATGTC	
CsSnRK2.9	AGATTGTGAGGGAGGCAAGAAAGC	GTCGTGATCCTCTTCGGTGTTCG	
CsSnRK2.10	AACTCAGCCGCCTCGCAGTC	CCTCCGGTGCTGTGTATGTTGC	
CsSnRK2.11	CAGACCGAAGAAGTTGTTGCCATG	ACCAAAGCCGCGTGGGTTTTG	
CsSnRK2.12	GCCCAACCATCAAAACCCATCATG	TCCTCTTGTTCCAGCCCAGTCTC	
actin	GCCCTCCCTCATGCCATTCT	TCGGCAGTGGTGGTGAACAT	

Results

Chromosome distribution and basic information of CsSnRK2 genes

To explore the function of the SnRK2 gene, we systematically analyzed the physicochemical properties of the cucumber SnRK2 protein. We identified 12 CsSnRK2 genes and named them sequentially from CsSnRK2.1 to CsSnRK2.12. The SnRK2 genes differed substantially according to the encoded protein size and biophysical properties (Table 2). The number of amino acids ranged from 298 to 673. The sequence of CsSnRK2.6 protein was the longest, containing 673 amino acids, while the sequence of CsSnRK 2.9 protein was the shortest, containing 298 amino acids. The molecular weight of CsSnRK2 ranged from 34,754.37 to 77,009.99 Da, and the theoretical isoelectric point (pI) ranged from 4.71 to 8.8. CsSnRK 2.5 and CsSnRK 2.7 were basic proteins, whereas the others were acidic proteins. The total average hydrophobic index of the 12 CsSnRK2 gene family members was negative, indicating that they were hydrophilic proteins.

Table 2 CsSnRK2 genes family members and characteristics.

Gene ID	Gene name	Chromosom	Gene location	Amino acid number	Molecular weight/Da	pI	Instability index	Aliphatic index	Grand average of hydropathicity	
			Start position	End position							
CsaV3_2G014200.1	CsSnRK2.1	2	11843438	11848872	361	41238.84	4.71	53.36	89.89	−0.301	
CsaV3_2G016930.1	CsSnRK2.2	2	14181637	14188320	363	41074.59	4.68	36.96	87.02	−0.286	
CsaV3_3G002220.1	CsSnRK2.3	3	1788904	1793145	340	38382.87	5.76	35.92	90	−0.306	
CsaV3_4G030050.1	CsSnRK2.4	4	19803919	19811002	355	40895.4	5.99	45.06	79.61	−0.565	
CsaV3_4G030060.1	CsSnRK2.5	4	19819061	19836491	543	62061.34	7.79	47.81	84.2	−0.456	
CsaV3_4G030080.1	CsSnRK2.6	4	19839540	19846552	673	77009.99	6.56	46.78	85.88	−0.423	
CsaV3_4G030100.1	CsSnRK2.7	4	19856029	19860650	372	42436.16	8.8	40.48	77.82	−0.558	
CsaV3_4G030120.1	CsSnRK2.8	4	19871598	19886067	486	55726.63	5.53	46.09	83.6	−0.409	
CsaV3_4G030140.1	CsSnRK2.9	4	19893248	19896492	298	34754.37	6.09	52.65	71.98	−0.712	
CsaV3_4G030260.1	CsSnRK2.10	4	19973756	19978649	344	38949.36	5.24	36.12	90.67	−0.293	
CsaV3_6G045260.1	CsSnRK2.11	6	26819227	26822605	355	40935.31	5.67	45.71	79.61	−0.579	
CsaV3_7G008840.1	CsSnRK2.12	7	5514315	5517582	365	41222.74	4.77	37.92	90.55	−0.314	

Subcellular localization prediction (Table 3) results showed that all members of the cucumber SnRK2 gene family were localized in the cytoplasm, and most were also localized in the nucleus. The predicted secondary structure of the cucumber SnRK2 protein was mainly composed of an α-helix (36.05%–44.97%) and irregular curl (36.10%–45.43%), and the extended structure (9.40%–20.99%) and ß-turn (3.76%−7.37%) were relatively small.

Table 3 Secondary structure and subcellular localization of CsSnRK2 genes.

Gene ID	Gene name	α-helix %	Beta turn %	Random coil %	Extended strand %	Subcellular localization	
CsaV3_2G014200.1	CsSnRK2.1	38.50	5.54	38.50	17.45	Cytoplasmic, Cytoskeleton	
CsaV3_2G016930.1	CsSnRK2.2	36.09	6.61	41.32	15.98	Cytoplasmic, Chloroplast	
CsaV3_3G002220.1	CsSnRK2.3	39.41	4.41	40.29	15.88	Cytoplasmic, Nuclear	
CsaV3_4G030050.1	CsSnRK2.4	41.41	5.35	38.03	15.21	Cytoplasmic, Cytoskeleton	
CsaV3_4G030060.1	CsSnRK2.5	35.54	7.37	36.10	20.99	Cytoplasmic, Nuclear	
CsaV3_4G030080.1	CsSnRK2.6	41.60	4.90	36.85	16.64	Cytoplasmic, Nuclear	
CsaV3_4G030100.1	CsSnRK2.7	34.41	3.76	45.43	16.40	Cytoplasmic, Nuclear	
CsaV3_4G030120.1	CsSnRK2.8	41.15	3.70	39.71	15.43	Cytoplasmic, Nuclear	
CsaV3_4G030140.1	CsSnRK2.9	44.97	3.69	41.95	9.40	Cytoplasmic, Nuclear	
CsaV3_4G030260.1	CsSnRK2.10	43.38	6.20	35.21	15.21	Cytoplasmic, Cytoskeleton	
CsaV3_6G045260.1	CsSnRK2.11	36.05	5.52	41.57	16.86	Cytoplasmic, Nuclear	
CsaV3_7G008840.1	CsSnRK2.12	36.44	6.03	40.55	16.99	Cytoplasmic, Chloroplast	

Chromosome analysis of the cucumber SnRK2 gene family showed that 12 CsSnRK2 genes were unevenly distributed on five cucumber chromosomes (Fig. 1). There were two genes distributed on chromosome 2 (CsSnRK2.1 and CsSnRK2.2) and one gene each on chromosomes 3 (CsSnRK2.3), 6 (CsSnRK2.11), and 7 (CsSnRK2.12). The other genes were located on chromosome 4 (CsSnRK2.4-CsSnRK2.10). No members of this gene family were found on chromosomes 1 or 5.

Figure 1 Distribution of CsSnRK2 genes family on chromosome.

Chrom01-07 are chromosome names. The chromosome scale on the left is in millions of bases (Mb).

Collinearity and selective pressure analysis of the CsSnRK2 genes

We explored the evolutionary process of CsSnRK2, AtSnRK2, ZmSnRK2, and OsSAPK2 genes. Collinearity analysis using MCScanx was performed using the TBtools software (Fig. 2). There were three pairs of collinearity genes between cucumber and A.thaliana, four pairs between cucumber and rice, and one pair between cucumber and maize. The col-linearity of the AtSnRK2, OsSAPK, ZmSnRK2, and CsSnRK2 genes could be broadly classified into two categories; one-to-many (CsSnRK2.11-AtSnRK2.7/8, CsSnRK2.11-OsSAPK1/2), or one-to-one (CsSnRK2.2-AtSnRK2.2, CsSnRK2.12-OsSAPK8, CsSnRK2.3-OsSAPK3, CsSnRK2.12- ZmSnRK2.8). In addition, these gene pairs belonged to the same group in the phylogenetic tree (Fig. 3), and their gene structure and conserved motifs were highly similar (Fig. 4).

Figure 2 Collinearity analysis of cucumber with Arabidopsis, rice and maize.

The gray rectangles represent chromosomes. The chromosome of cucumber (Cs-Chr1-7) is shown in red. Arabidopsis chromosomes (At-Chr1-5), rice chromosomes (Os-Chr1-12) and maize chromosomes (Zm-Chr1-10) are shown in blue. The gray lines represent collinear blocks of the SnRK2 gene, and the colored lines represent collinear pairs of the SnRK2 gene.

Figure 3 Phylogenetic analysis of the SnRK2s family genes from cucumber, A.thaliana, rice, maize, sugar beet, tomato, grapevine, and pak-choi.

Royal blue star represent cucumber, purple circles represent A.thaliana, dark gray circles represent rice, yellow circles represent maize, dark cyan circles represent sugar beet, blue circles represent tomato, pink circles represent grapevine, orange circles represent pak-choi. phylogenetic tree was constructed using the minimum neighbor method by MEGA 11. The bootstrap values of 1000 replicates were calculated at each node. Lines and arcs with different colors represent different subfamilies.

Figure 4 Phylogenetic tree, structure and conserved motif relationship of CsSnRK2 and AtSnRK2 genes.

(A) The SnRK2 phylogenetic tree is divided into three groups (I. II, and III), and different colors represent different branches. (B) The blue boxes, black lines, and yellow ellipse in the gene structural diagram represent untranslated region, introns, and CDS sequence, respectively. (C) Analysis with MEME to investigate 10 conserved motifs of SnRK2 proteins. Motifs are indicated by 10 different colored boxes. (D) Basic composition of 10 conserved motifs.

Purifying selection is the process of removing unfavorable mutations, whereas positive selection is the accumulation of new favorable mutations and their transmission to the entire population (Qiao et al., 2015). The value of dN/dS plays a crucial role in gene selection and evolution; dN/dS >1 is positive selection, dN/dS = 1 is neutral selection, 0 < dN/dS <1 is purifying selection (Yadav et al., 2015). The results showed that all three pairs of genes had dN/dS values <1; indicating that the evolutionary process underwent purifying selection pressure and evolved conservatively, which helped maintain the basic functions of the genes (Table 4).

Table 4 Selective pressure analysis of CsSnRK2 genes family.

A pair of genes	S	N	d S	d N	d N /d S	
AtSnRK2.2∼CsSnRK2.2	254.2	768.8	3.5115	0.1588	0.0452	
AtSnRK2.7∼CsSnRK2.11	235.8	766.2	4.4248	0.1547	0.0350	
AtSnRK2.8∼CsSnRK2.11	241.1	841.9	2.2890	0.0930	0.0406	
OsSAPK1∼CsSnRK2.11	247.7	775.3	5.7825	0.1281	0.0222	
OsSAPK2∼CsSnRK2.11	199.0	590.0	6.7995	0.1378	0.0203	
OsSAPK3∼CsSnRK2.3	250.4	751.6	4.7973	0.1483	0.0309	
OsSAPK8∼CsSnRK2.12	242.7	807.3	6.9083	0.0831	0.0120	
ZmSnRK2.8∼CsSnRK2.12	265.8	817.2	11.7553	0.0894	0.0076	
Notes.

* “S” represents the numbers of synonymous, “N” represents non-synonymous, “dS” represents the synonymous mutation fre-quency and “dN” represents the non-synonymous mutation frequency and the ratio of dN/dS.

Phylogenetic tree analysis of the CsSnRK2 genes

To explore the functional characteristics and evolutionary relationships of the CsSnRK2 genes, a phylogenetic tree was constructed using multiple sequences alignments of SnRK2 protein sequences from the 12 cucumber, 10 Arabidopsis, 10 rice, 11 maize, six sugar beet, seven tomato, eight grapevine, and 13 pak-choi genes. The phylogenetic tree showed that the SnRK2 gene family was clustered into three branches, labeled as group I, group II, and group III (Fig. 3), which is consistent with the results of previous studies in Arabidopsis (Umezawa et al., 2009). The cucumber SnRK2 family members of cucumbers were unevenly distributed among the three groups. Group I had eight members, group II had one member, and group III had three members. The phylogenetic tree showed that CsSnRK2 was more closely grouped with SnRK2 from the dicots Arabidopsis, sugar beet, tomato, grapevine, and pak-choi than with SnRK2 from the monocots rice and maize. The cucumber SnRK2 and grapevine SnRK2 proteins were highly homologous, indicating that they have a close phylogenetic relationship and conservative evolution. Specifically, CsSnRK2.3 and VvSnRK2.6; CsSnRK2.11 and VvSnRK2.4/8; and CsSnRK2.2 and VvSnRK2.1 had the closest phylogenetic relationships, indicating that they may have the most similar functional characteristics.

Structure and conserved motif analysis of the CsSnRK2 genes

Analysis of the gene structure and conserved motifs of cucumber SnRK2 provided important information for understanding the evolution of this gene (Fig. 4). Therefore, a phylogenetic tree was constructed using the CsSnRK2 and AtSnRK2 genes. According to the phylogenetic tree, the cucumber and Arabidopsis SnRK2 gene families could be divided into three groups (Fig. 4A). In the third group, all cucumber genes differed from the previous (Fig. 3). The exon-intron distribution of cucumber SnRK2 genes was conserved, most of these had nine exons and eight introns; this was consistent with most Arabidopsis SnRK2 genes, except for AtSnRK2.6 (10 exons), AtSnRK2.3 (10 exons), and AtSnRK2.8 (6 exons) (Fig. 4B) (Yoon et al., 1997). In addition, the number and location of introns and exons in the third group of CsSnRK2 were different from those in Arabidopsis homologs, which may contribute to the functional diversification of CsSnRK2. Moreover, most introns of the cucumber SnRK2 gene were found to be much longer than those in the Arabidopsis SnRK2 genes.

We used TBtools to predict the conserved motifs in the CsSnRK2 and AtSnRK2 families (Fig. 3C). A total of 10 conserved motifs were identified, and their motif sequences are shown in Fig. 4D. Except for AtSnRK2.3, all genes had motifs 1-10, and the sequence and position of the motif in the structure were nearly the same, showing high conservatism in evolution. In the third group, there were significant differences in motif placement and location, and motif deletions were also observed. For example, CsSnRK2.7 and CsSnRK2.8 lacked motif 9, while CsSnRK2.9 lacked motif 4/5/6 but hads two of motif 1, which was similar to that of CsSnRK2.5. We hypothesized that these differences were associated with gene-specific functions.

Cis-acting element of the CsSnRK2 promoter region

The plant SnRK2 gene family plays an important role in abiotic stress responses, and the analysis of regulatory elements provides favorable information for the study of gene function. Several cis elements were identified in the promoter regions of the CsSnRK2 genes (Fig. 5). Among them, at large variety of hormone-related cis elements, such as P-box, TATC-box, GARE-motif, TCA-element, ABRE, TGACG-motif, CGTCA-motif, TGA-element, and AuxRR-core, indicated that they responded to different hormones such as MeJA, growth hormone, ABA, ethylene, GA and SA. The four cis elements were MBS, LTR, TC-rich repeats and ARE, which are involved in drought, low temperature, defense, stress and anaerobic induction. In addition, six cis elements associated with development were identified in the CsSnRK2 promoter, including P-box, TATC-box, Gare-Motif, ABRE, and TGA-element/AuxRR-core. The CsSnRK2.8 gene had the least active elements, containing only three elements, while the other genes had at least five elements. The genes, except CsSnRK2.7 and CsSnRK2.8, contained ABRE elements. This suggests that most CsSnRK2 genes are responsive to ABA. Only four members (CsSnRK2.1, CsSnRK2.2, CsSnRK2.4, and CsSnRK2.7) contained the LTR-element. In addition, two members (CsSnRK2.9, and CsSnRK2.11) contained TC-rich repeats, and five members (CsSnRK2.3, CsSnRK2.7, CsSnRK2.8, CsSnRK2.11, and CsSnRK2.12) contained MBS. We hypothesized that exogenous stress could induce CsSnRK2 gene expression through cis-responsive elements, thereby enhancing plant resistance to adversity.

Figure 5 The 2 kb upstream region cis-acting element of SnRK2 gene family in cucumber.

(A) The distribution of cis-acting elements in the promoter sequence of CsSnRK2 genes are represented by different colors. (B) Number of cis-acting elements statistics. (C) Cis-element names and their functional notes.

Tissue-specific expression analysis of the CsSnRK2 genes

The heat map of RNA-Seq data expression of the CsSnRK2 genes in different tissues and organs using TBtools software (Fig. 6) revealed that CsSnRK2.1 and CsSnRK2.6 were not expressed in almost all tissues. CsSnRK2.2, CsSnRK2.4, CsSnRK2.8, CsSnRK2.10 and CsSnRK2.12 were strongly expressed in each tissue, and their function may be related to growth and development. CsSnRK2.5 was strongly expressed in the roots and moderately expressed in unfertilized ovaries. CsSnRK2.7 was strongly expressed in the roots and weakly expressed in other tissues. It was speculated that CsSnRK2.5 and CsSnRK2.7 may play an important role in root growth. CsSnRK2.11 was highly expressed in tendrils and the base of tendrils, and moderately expressed in other tissues. CsSnRK2.3 was moderately expressed in all the tissues. CsSnRK2.9 was moderately expressed in the roots and weakly expressed in the leaves.

Figure 6 Heat map of tissue-specific RNA-Seq data expression of SnRK2 genes in Cucumber.

The CsSnRK2 genes were listed on the right side of the expression array, and the color bar to the right of the genes are a scaled log2 expression value based on RNA-Seq data, indicating a color gradient from low expression (light blue) to high expression (black).

qRT-PCR expression characteristics of the CsSnRK2 genes

The expression levels of the CsSnRK2 genes under salt, PEG, and ABA stress were determined using qRT-PCR. A total of 50 µmol/L ABA (T1), 100 µmol/L ABA (T2), 200 mmol/L NaCl (T3), and 10% PEG (T4) were sampled at 3, 6, 12, and 24 h, respectively, and 0 h was used as a control. The expression of CsSnRK2 was significantly different between the different time points and treatments (Fig. 7).

Figure 7 Real-time fluorescence quantitative expression analysis of CsSnRK2 genes.

(A) A total of 50 mmol/L ABA, (B) 100 mmol/L ABA, (C) 200 mmol/L NaCl, and (d) 10% PEG. The legend numbers 3 h, 6 h, 12 h and 24 h indicate the sampling time, and 0 h is set as the control. With CsActin as the internal parameter, the relative expression was calculated by 2- ΔΔCt method, and value represents mean ±SE of the three biological replications. Asterisks indicated values that are significantly different from CK (0 h) (* p < 0.05, ** p < 0.01, one-way ANOVA).

Under 50 µmol/L ABA treatment (Fig. 7A), four genes (CsSnRK2.4/5/11/12) were up-regulated, of which CsSnRK2.11, and CsSnRK2.12 were up-regulated the most, and their relative expression levels at 6 and 24 h were 4 and 3.5 times higher than that at 0 h, respectively. CsSnRK2.4 was highly expressed at 24 h, and its relative expression level at 24 h was 4.3 times higher than that at 0 h. In addition, five genes (CsSnRK2.2/6/8/9/10) were down-regulated, and CsSnRK2.1/3/7 were up-regulated first and then down-regulated. CsSnRK2.3/7 genes reached their maximum expression values at 3 h.

Upon 100 µmol/L ABA treatment (Fig. 7B), all the genes were up-regulated and then down-regulated, except for CsSnRK2.2 and CsSnRK2.5, which were downregulated. All three genes (CsSnRK2.4/11/12) were up-regulated, of which CsSnRK2.11 was up-regulated the most, and the relative expression level at 24 h was 11.9 times higher than that at 0 h. The expression of CsSnRK2.1 was the highest at 6 h, and the relative expression levels at 6 h was 6.8 times higher than that at 0 h. The maximum expression of CsSnRK2.2/6/7/9/10 was reached after 3 h; however, the expression of CsSnRK2.2 was down-regulated by both ABA treatments, suggesting that CsSnRK2.2 may play a negative regulatory role.

In response to 200 mmol/L NaCl treatment (Fig. 7C), three genes (CsSnRK2.4/11/12) were up-regulated, of which CsSnRK2.11 and CsSnRK2.12 were up-regulated the most; their relative expression levels at 12 and 24 h were 7.3 and 5 times higher than that at 0 h, respectively. In addition, five genes (CsSnRK2.2/6/8/9/10) were down regulated. The genes CsSnRK2.1 and CsSnRK2.7 were up-regulated first and then down-regulated.

For the 10% PEG treatment (Fig. 7D), six genes (CsSnRK2.1/5/7/10/11/12) were up-regulated, where CsSnRK2.11 and CsSnRK2.12 were up-regulated the most; their relative expression level at 12 and 24 h were 15.7 and 9.6 times higher, respectively, that of 0 h. However, two genes (CsSnRK2.2/4) were also down-regulated. The expression of CsSnRK2.5 reached a maximum at 12 h, and its relative expression level at 12 h was 6.6 times higher than that at 0 h.

This study provides an overview of CsSnRK2 expression under ABA, NaCl, and PEG stress conditions, where different CsSnRK2 members had different expression patterns under these stressors. This illustrates that the functional diversity of this gene is widely present in different plant species.

Analysis of the protein interaction network of the CsSnRK2 family genes

A protein–protein interaction network between CsSnRK2 and other cucumber proteins was constructed using the STRING database. According to the prediction results, cucumber SnRK2 protein can interact with five other proteins (XP_004146814.1, XP_004148120.1, XP_004167207.1, XP_004154794.1, and XP_004168589.1), the first four of which are PP2C proteins, which belong to the PP2C superfamily, and the fifth is a sucrose non-fermentable protein, presumably belonging to the other subfamilies outside SnRK2. The results also showed that eight CsSnRK2 proteins interacted with XP_004146814.1 (PP2C2), XP_004148120.1, and XP_004167207.1. The XP_004154794.1 protein interacted with five CsSnRK2 proteins. In addition, no interaction between CsSnRK2 proteins were found in the prediction results; however, the CsSnRK2 protein interacted with a sucrose non-fermenting protein (XP_004168589.1). We speculated that this protein belongs to a protein family other than SnRK2 and plays an important role. These results provide useful information for further analysis to verify the function of CsSnRK2 genes.

Discussion

Plants evolve and develop with changes in the environment, gradually forming a self-protection mechanism against abiotic stress. Under adverse conditions, this protective mechanism can activate the expression of relevant genes and change the structure of relevant functional proteins to protect the normal metabolic response in cells (Bohnert et al., 2006). The plant hormone pathway plays a key role in the ability of plants to adapt to stressful environment (Fujita et al., 2006; Verma, Ravindran & Kumar, 2016). ABA is an important plant hormone with many physiological functions in modulating plant growth and development, salt stress, water deficit, and other stress responses (Golldack et al., 2013; Zhang et al., 2006). In these processes, stress-induced phosphorylation of protein kinases plays a key role in plant perception and response to the environment (Fujita et al., 2006; Wei et al., 2017). SnRK2 is a plant-specific serine/threonine kinase that is activated in response to biotic and abiotic stresses in plants (Coello, Hey & Halford, 2011).

The SnRK2 gene family has been identified in many species, but only a few genes have been functionally validated. Comprehensive and systematic studies on the Cucurbitaceae are limited; thus, the analysis of CsSnRK2 gene function and characterization deserves specific attention. In this study, 12 CsSnRK2 genes were identified from the cucumber genome database and named CsSnRK2.1-CsSnRK2.12 (Table 1). Chromosome distribution and protein physical and chemical property analysis revealed that 12 SnRK2 genes were unevenly distributed on five cucumber chromosomes (chromosomes 2/3/4/6/7), and most of genes were distributed on chromosome 4 (Fig. 1). The three main causes of gene family expansion are tandem replication, fragment replication, and genome-wide replication, which may also produce variations in genetic material that are better adapted to the pressures of natural selection during evolution (Cronk, 2001; Wei, Wang & Xie, 2014). BLAST using MCScanx provided by TBtools revealed no collinear relationship between CsSnRK2 genes; however, the presence of tandem duplication in multiple genes on chromosome 4 remains to be verified. In addition, the collinearity between A.thaliana, maize, rice, and cucumber was analyzed (Fig. 2). There were three pairs of collinear genes between cucumber and A.thaliana, four pairs with rice, and one pair with maize. We hypothesized that these genes may be derived from a common ancestor and contribute to the cucumber SnRK2 gene family. Further analysis of the role of evolutionary constraints showed that the dN/dS values of all three gene pairs were <1 (Table 4), indicating that the evolutionary process underwent purification selection pressure and was highly conservative, which helped maintain the basic functions of the genes.

Phylogenetic analysis and comparisons were used to understand the evolutionary relationships between genes and species. Cucumber SnRK2 genes could be divided into three groups (Fig. 3), which is consistent with studies on other species, except for soybean (Wei et al., 2017). CsSnRK2 had high homology with the protein of dicotyledon SnRK2 and was far away from monocotyledons; collinearity analysis showed that CsSnRK2 had relatively few collinear gene pairs with monocotyledon maize and rice and dicotyledon Arabidopsis. These results suggest that SnRK2 may have evolved from a common ancestor and rapidly diverged after the segregation of monocotyledons and dicotyledons. This further indicated that CsSnRK2 genes have a relatively conserved evolutionary process.

The structure, length, and number of introns and exons are important features and traces of the evolution of certain gene families (Chen et al., 2013). According to our analysis, the number and size of SnRK2 exons were highly conserved between cucumber and Arabidopsis, with some exceptions. Structural analysis of CsSnRK2 genes showed two patterns, most of which had nine CDS and eight introns (Fig. 4B). The former gene structure pattern was consistent with that of cotton, and rice (Liu et al., 2017; Saha et al., 2014); however the exons and introns of CsSnRK2 genes in the third group were different from those in the first and second groups, which may be related to the functional diversity of SnRK2 genes in cucumber. Previous studies have shown that genes with fewer introns are more highly expressed (Chung et al., 2006). In the present study, CsSnRK2 gene expression levels were higher in those with fewer introns, which is consistent with previous studies. In addition, the length of CsSnRK2 gene introns was generally higher than that of Arabidopsis; therefore, we speculated that the introns of the CsSnRK2 genes may be the imprinting that led to the evolutionary process and functional diversity of this gene family.

The SnRK2 family has a highly conserved N-terminal structural domain and C-terminal regulatory structural domain, and the functional diversity of the SnRK2 gene is closely related to its C-terminus (Kulik et al., 2011). The C-terminal domain activates related kinases through protein interactions in response to ABA and signaling (Vlad et al., 2009). In our study (Fig. 4C), we found that groups I and II contained motifs 1–10, except for AtSnRK2.7 which did not contain motif 9. In addition, motif 9 was missing in the third group CsSnRK2.7 and CsSnRK2.8, while CsSnRK2.9 was missing motif 4/5/6. Previous reports have demonstrated that there are differences in gene structure and physical and chemical properties among gene families, which may be due to the structural and property changes of gene families to better adapt to the environment during the evolutionary process (Jaramillo & Kramer, 2007; Thornton Joseph & De Salle 2000). Therefore, we hypothesized that motifs 1/2/3/7/10 are important components of the C-terminus of the amino acid sequence of the CsSnRK2 gene family. A few gene motif deletions may be due to structural polymorphisms in SnRK2 during long-term evolution, different evolutionary pathways, and some functional divergence.

When plants are stressed, through a series of signal transductions, the plants themselves produce resistance factors to activate the related transcription factors in the plant body and corresponding cis-regulatory elements and activate gene expression to respond to stress (Ali & Komatsu, 2006; Hadiarto & Tran, 2011). A variety of cis-regulatory elements were found in the promoter region of the CsSnRK2 gene in this study (Fig. 5). A previous study has shown that the combination of multiple ABRE, or one ABRE and the corresponding DRE/CRT elements, can lead to the expression of ABA genes (Fujita, Yoshida & Yamaguchi-Shinozaki, 2013). Our results showed that all genes, except CsSnRK2.7 and CsSnRK2.8, contained ABRE (Fig. 4). However, previous studies have shown that not all genes induced by specific stressors have corresponding cis-acting elements in their promoters (Zhang et al., 2016). For instance, the CsSnRK2.7 and CsSnRK2.8 promoters, which do not contain ABRE, could also be induced by ABA (Fig. 6). These results suggest that there may be other unknown stress-related cis-acting elements or mechanisms involved in the regulation of these genes.

Studies have shown that in Arabidopsis, the first group of SnRK2B is expressed in a variety of tissues (Fujii, Verslues & Zhu, 2007). GmSnRK2.14 and GmSnRK2.15 were highly expressed in late soybean seed development, and GmSnRK2.2 and GmSnRK2.16 were preferentially expressed in meristematic tissues (Wei et al., 2017). We analyzed the expression levels of the CsSnRK2 gene family in different tissues using online transcriptome RNA-Seq data (Fig. 6). CsSnRK2 showed a specific expression pattern in each group. CsSnRK2.5 and CsSnRK2.7 were highly expressed in the roots and CsSnRK2.11 was highly expressed in tendrils and tendril bases. The expression of CsSnRK2.1, CsSnRK2.6, and CsSnRK2.7 in the first group was almost undetectable in all tissues. CsSnRK2.2/4/8/10/12 were highly expressed in all tissues, and their function may have an important relationship with growth and development. In our gene expression analysis, CsSnRK2 expression in the leaf tissues differed from that in the transcriptome RNA-Seq data. For example, CsSnRK2.1 was induced by ABA, PEG, and NaCl to upregulate its expression. CsSnRK2.8 has only three elements, LTR, P-box, and MBS, which are involved in low temperature, gibberellin, and drought-inducible elements, respectively; however, CsSnRK2.8 was moderate to highly expressed in various tissues (Fig. 6). These results suggest that the activity of the CsSnRK2 genes is not specifically associated with region differences.

Previous studies have shown that SnRK2 genes in Arabidopsis can be clustered into three groups. Groups I and II are activated under hyperosmotic stress, and members of groups II and III responded to both ABA and hyperosmotic signals (Nakashima et al., 2009). In particular, the SnRK2 gene family in the third group was highly induced by ABA and is an important component in the response to ABA signaling (Yamaguchi-Shinozaki, 2010; Yoshida et al., 2006). In the present study, CsSnRK2.1 and CsSnRK2.12 in the third group were strongly induced by ABA (Figs. 7A, 7B). In the second group, CsSnRK2.11 was strongly induced by ABA. The third group of CsSnRK2 showed a weak ABA response. These results are not entirely consistent with those of previous studies in Arabidopsis, where the group III members AtSnRK2.2, AtSnRK2.3, and AtSnRK2.6, were strongly induced by ABA (Kobayashi et al., 2004). Some SnRK2 genes are the main regulatory factors in ABA signal transduction, but some may not be affected by ABA and have acquired unique regulatory properties (Boudsocq, Barbier-Brygoo & Lauriere, 2004; Kobayashi et al., 2004). Thus, the fact that the third group of CsSnRK2.2 was not activated by ABA, indicates a specific role of CsSnRK2.2 in response to ABA signaling. All TaSnRK2 members are rapidly induced by PEG and NaCl treatments (Zhang et al., 2016). In this study, CsSnRK2.11 and CsSnRK2.12 of the second and third groups were strongly induced by PEG and NaCl (Figs. 7C, 7D). Moreover, the elements involved in the drought-induced response (MBS) and the cis-acting elements (TC-rich repeats) involved in defense and stress response were also found in the promoter, indicating that these elements may be responsible for the co-expression of CsSnRK2 in the stress response of cucumber under different environmental conditions (Fig. 4). The first group of CsSnRK2 was weakly responsive to PEG and NaCl treatments, except for CsSnRK2.4 and CsSnRK2.5 which were strongly activated by NaCl and PEG, respectively. The third group of CsSnRK2.2 was not activated by drought and salt stresses, suggesting that CsSnRK2.2 plays a special role in responding to hyperosmotic signals. Therefore, these results are not entirely consistent with the other Arabidopsis studies (Saha et al., 2014). Taken together, the expression profile of CsSnRK2 under abiotic stress and ABA treatment suggests that SnRK2 genes have conserved and diverse biological functions in the plant kingdom.

PP2C and SnRK2 are second and third messengers of the ABA signal transduction pathway. The inactivation of PP2C phosphatase allows SnRK2 inhibition to be lifted, activating the phosphorylation mechanism and catalyzing downstream transcription factors that regulate plant resistance to adversity (Ma et al., 2009; Soon et al., 2012). Studies on rice OsPYL3 have revealed that OsPYL3 can interact with some class A OsPP2C genes and that transgenic rice seeds have enhanced tolerance to cold and drought stresses during germination (Tian et al., 2015). In Arabidopsis, type A PP2C of regulates ABF, SLAC1, and other components in the downstream signaling pathway, by inhibiting the activity of AtSnRK2.2, AtSnRK2.3 and AtSnRK2.6 protein kinases; thus mediating the response to stress (Fujita et al., 2009). In one study, three CsPYL, four CsPP2C, and two CsSnRK2 genes were obtained by homologous cloning. Phylogenetic tree analysis of Arabidopsis and cucumber genes showed that CsPP2C2 belonged to group A PP2C and CsSnRK2.2 belonged to group III, and these two genes were hypothesized to be involved in ABA signal transduction (Wang et al., 2012). In our prediction (Fig. 8), multiple CsSnRK2 proteins interacted with four proteins, including XP_004146814.1 (PP2C2) and another PP2C, and all of them responded to ABA treatment to varying degrees (Fig. 7); for example, CsSnRK2.11/12 responded strongly to ABA. This suggests that the CsSnRK2 genes may also play an important role as a third messenger in the ABA signaling pathway. In addition, sucrose non-fermentable proteins (XP_004168589.1) which we speculate may belong to a subfamily, other than SnRK2, interact with the CsSnRK2 protein, and play important roles in some specific regulatory pathways.

Figure 8 Interaction network analysis of CsSnRK2 protein.

Protein–protein interaction networks were predicted using STRING. Each node represents a protein, and each edge represents an interaction, colored by evidence type.

Conclusions

In this study, in a genome-wide search of cucumber, we identified 12 SnRK2 gene family members and analyzed them in detail. These SnRK2 genes were distributed on five chromosomes, and phylogenetic clustering divided them into three well-supported clades. In addition, collinearity analysis showed that the CsSnRK2 gene family has underwent purifying selection pressure during evolution. CsSnRK2 genes of the same group had similar exons and conserved motifs, and their intron length may be a specific imprint for the evolutionary amplification of the CsSnRK2 gene family. By predicting cis elements in the promoter, we found that the promoter region of CsSnRK2 gene members had various cis-regulatory elements in response to hormones and stress. Relative expression analysis showed that CsSnRK2.11 (group II) and CsSnRK2.12 (group III) was strongly induced by ABA, NaCl and PEG stress, respectively; whereas CsSnRK2.2 (group III) was not activated by any treatment. The response of group I CsSnRK2 to ABA, NaCl and PEG was weak. Protein interaction prediction showed that multiple CsSnRK2 proteins interacted with four proteins, including PP2C, and the CsSnRK2 genes may also play an independent role as third messengers in the ABA signaling pathway. These results provide a reference for future analysis of the potential function of CsSnRK2 genes.

Supplemental Information

Supplemental Information 1 Protein sequences

Click here for additional data file.

Supplemental Information 2 The original CT value of qRT-PCR

Click here for additional data file.

Additional Information and Declarations

Competing Interests

Author Contributions

Data Availability

The authors declare there are no competing interests.

Zilong Wan performed the experiments, analyzed the data, prepared figures and/or tables, and approved the final draft.

Shilei Luo conceived and designed the experiments, prepared figures and/or tables, authored or reviewed drafts of the article, and approved the final draft.

Zeyu Zhang performed the experiments, analyzed the data, prepared figures and/or tables, and approved the final draft.

Zeci Liu analyzed the data, prepared figures and/or tables, authored or reviewed drafts of the article, and approved the final draft.

Yali Qiao analyzed the data, prepared figures and/or tables, and approved the final draft.

Xueqin Gao analyzed the data, prepared figures and/or tables, and approved the final draft.

Jihua Yu analyzed the data, prepared figures and/or tables, and approved the final draft.

Guobin Zhang conceived and designed the experiments, prepared figures and/or tables, authored or reviewed drafts of the article, and approved the final draft.

The following information was supplied regarding data availability:

The raw data is available in the Supplementary File.

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
