# Peer review of "Identification and expression profile analysis of the SnRK2 gene family in cucumber"

_PeerJ, doi:10.7717/peerj.13994_

## Round 0.1 · original submission · Major Revisions

The peer review process identified a number of comments or suggestions for revision. Please evaluate the comments carefully and if you feel you can address the issues, we would welcome a revision. A revision will be sent out again for review, but I cannot guarantee it will be accepted for publication.

Reviewer 1 ·

Basic reporting

The manuscript requires English editing. The are plenty of syntax errors, typos and grammar flubs throughout the manuscript.
The authors use the word "respectively" without making syntax sense. eg. lines 26, 57, 85, 164, 165 etc
The legend of most of the figures lacks information and they do not describe the figure adequately.

Experimental design

The authors have found (through bioinformatic analysis) and studied 12 SnRK2 in cucumber genome. They have also found that CsSnRK2 gene family has possibly been under selection pressure during evolution. They have also used previously published RNASeq data to study the expression patterns of CsSnRK2 family in several cucumber tissues.
The methods look fine to me although they do not describe in details the parameters and filtering criteria that they used for the several bioinformatic tools.

Validity of the findings

Conclusions and discussion are well stated.

qPCR results without statistics on Figure 7 (eg. ttest, number of biological replicates etc) and what was the housekeeping gene?

Reviewer 2 ·

Basic reporting

The manuscript by Wan et al reports the identification by in-silico analysis and gene studies of cucumber SnRK2 genes. In general, the manuscript is well structured, research question is adequately defined. The whole manuscript is easy to follow, results are clearly presented, literature references are sufficient and raw data is shared.

Experimental design

The manuscript by Wan et al reports the identification by in-silico analysis and gene studies of cucumber SnRK2 genes. In general, the manuscript is well structured, research question is adequately defined, and research provides new information that could fill a knowledge gap. The whole manuscript is easy to follow, results are clearly presented however there seems to be little experimental data to support the research hypothesis. For example, since there is interaction between CsSnRK2 protein family and other protein families, authors could use additional molecular techniques to verify this interaction and accomplish the rigorous investigation of the subject.
However, the most important issue is the lack of essential information and adequate description of Materials and Methods. This section needs more detail, and it needs to be written in a more descriptive way since current writing includes short, “automated” phrases that do not help the comprehension of the manuscript nor its scientific presentation/replication. The most crucial matter is that there is essential information missing regarding the actual experiments. For example, how many seedlings were treated with each different treatment? And how many samples were taken from different time points/separate treatments? What is the statistical analysis of expression levels? For the latter, there is nothing mentioned on the corresponding section nor in the relevant figure.
The next important issue is the consistency in reporting the results. For example, Figure 2 includes the collinearity analysis of cucumber, Arabidopsis, rice and maize proteins. Yet in the manuscript, authors provide a description of analysis only for cucumber vs Arabidopsis proteins. In Figure 3 Group I has 3 member, Group II has 1 member and Group III has 8 members. That is not what’s written in the text (Lines 184-185). These inconsistencies should be corrected.

Validity of the findings

Underlying data have been provided however there is a lack of statistical analysis of gene expression (significance). Little experimental data to support the discussion/conclusions when more molecular analysis could have been used. Please provide more data if any.

Additional comments

Specific comments:
In phylogenetic tree analysis; the results are expected, dicots are surely more closely related than with monocots. Why didn’t authors include more sequences from dicots species, including the genes that are functionally characterized already?
The least important points have to do with language and again – consistency. For example,
Line 236; what does “may have some form of expression mean?”
Analysis of protein networks is about protein-protein interactions. Yet in Lines 280-291 genes get confused with proteins. It is the proteins that interact with each other, not the genes. Again, in Line 346 “the amino acid sequence of SnRK2 gene”, this is not right. I suggest you correct the whole manuscript accordingly.
Lines 384-387; Figure 7 includes the expression of SnRK2 genes in leaves under different stresses (according to M&M section). Why do you mention various tissues here?
Line 400; what is the third group of CsSnRK2.2? Probably is CsSNRK2.2 that belongs to Group III.
Line 425; why CsPP2C2?

---

## Round 0.2 · Minor Revisions

You are advised that the manuscript is not acceptable for publication at this time, but could be reconsidered for publication after minor revision.

Reviewer 1 ·

Basic reporting

no comment

Experimental design

no comment

Validity of the findings

no comment

Reviewer 2 ·

Basic reporting

English language needs revising/checking throughout the manuscript. There are still sentences describing methods in the M&M section where the verb is totally missing. See for example, sentences 93-95 and 99 to 102, 108-110, 110-111, 144-145. Also, sentences where the syntax or the meaning is complicated; for example line 127.

Experimental design

Lines 89-92 does that mean that authors had 4 samples (each consisting of two leaves) per treatment per time point? I still cannot figure out from the description of results how many samples were analyzed, i.e. how many samples were used for the average expression seen in the figure. Authors need to include that information in their manuscript.

Validity of the findings

no comment

---

## Round 0.3 · accepted · Accept

I am writing to inform you that your manuscript - Identification and expression profile analysis of the SnRK2 gene family in cucumber - has been Accepted for publication. Congratulations!